# Tying the knot: Unraveling the intricacies of the coronavirus frameshift pseudoknot

**Luke Trinity** [1]*, **Ulrike Stege**[1‡]*, **Hosna Jabbari**[2,3‡]*

**1** Department of Computer Science, University of Victoria, Victoria, British Columbia, Canada, **2** Department of Biomedical Engineering, University of Alberta, Edmonton, Alberta, Canada, **3** Institute on Aging and Lifelong Health, Victoria, British Columbia, Canada

‡ These authors are joint senior authors on this work.
* ltrinity@uvic.ca (LT); ustege@uvic.ca (US); jabbari@ualberta.ca (HJ)

**Data Availability Statement:** The datasets generated and/or analyzed during the current study are available in the repository: github.com/ltrinity/TyingTheKnot.

## Abstract

Understanding and targeting functional RNA structures towards treatment of coronavirus infection can help us to prepare for novel variants of SARS-CoV-2 (the virus causing COVID-19), and any other coronaviruses that could emerge via human-to-human transmission or potential zoonotic (inter-species) events. Leveraging the fact that all coronaviruses use a mechanism known as −1 programmed ribosomal frameshifting (−1 PRF) to replicate, we apply algorithms to predict the most energetically favourable secondary structures (each nucleotide involved in at most one pairing) that may be involved in regulating the −1 PRF event in coronaviruses, especially SARS-CoV-2. We compute previously unknown most stable structure predictions for the frameshift site of coronaviruses via hierarchical folding, a biologically motivated framework where initial non-crossing structure folds first, followed by subsequent, possibly crossing (pseudoknotted), structures. Using mutual information from 181 coronavirus sequences, in conjunction with the algorithm KnotAli, we compute secondary structure predictions for the frameshift site of different coronaviruses. We then utilize the Shapify algorithm to obtain most stable SARS-CoV-2 secondary structure predictions guided by frameshift sequence-specific and genome-wide experimental data. We build on our previous secondary structure investigation of the singular SARS-CoV-2 68 nt frameshift element sequence, by using Shapify to obtain predictions for 132 extended sequences and including covariation information. Previous investigations have not applied hierarchical folding to extended length SARS-CoV-2 frameshift sequences. By doing so, we simulate the effects of ribosome interaction with the frameshift site, providing insight to biological function. We contribute in-depth discussion to contextualize secondary structure dual-graph motifs for SARS-CoV-2, highlighting the energetic stability of the previously identified 3_8 motif alongside the known dominant 3_3 and 3_6 (native-type) −1 PRF structures. Using a combination of thermodynamic methods and sequence covariation, our novel predictions suggest function of the attenuator hairpin via previously unknown pseudoknotted base pairing. While certain initial RNA folding is consistent, other pseudoknotted base pairs form which indicate potential conformational switching between the two structures.

**Funding:** Funding provided by University of Victoria Department of Computer Science (https://www.uvic.ca/ecs/computerscience/index.php) to support LT. The Microsoft AI4Health Azure Grant (https://www.microsoft.com/en-us/ai/ai-for-health) to HJ enabled data collection via cloud computing. Finally, the Natural Sciences and Engineering Research Council (https://www.nserc-crsng.gc.ca/index_eng.asp) supported HJ, US, and LT. The funders had no role in study design, data collection and analysis, decision to publish, or preparation of the manuscript.

**Competing interests:** The authors have declared that no competing interests exist.

## Author summary

Finding evolutionary connections between coronaviruses frameshift element RNA structures is a worthwhile goal in contributing to treatment development for afflicted human and animal populations. Predicting the most energetically favourable RNA secondary structures, and how they may form via the hierarchical folding hypothesis, is an efficient use of computational resources to shed light on RNA structure-function.

We used the KnotAli algorithm to obtain mutual information from 181 coronaviruses frameshift RNA sequences. Guided by this evolutionary information, we computed secondary structure predictions to allow comparison of marked similarities and subtle differences between SARS-CoV-2 and other coronaviruses frameshift element RNA structures. In addition, we applied the Shapify algorithm to predict secondary structures for extended SARS-CoV-2 frameshift element sequences informed by SHAPE reactivity data. Here we critically expand the known landscape of most stable −1 PRF secondary structure conformations, isolating the location of key secondary structure motif transitions that can improve site targeting of viral therapeutics. Our application of hierarchical folding algorithms contributes novel predictions of functional RNA structures, enhancing discussion of how secondary structures unfold or refold to regulate frameshifting in coronaviruses.

## 1 Introduction

The virus SARS-CoV-2 (genus *Betacoronavirus*, subgenus *Sarbecovirus* i.e., Severe acute respiratory syndrome Beta-coronavirus [1]) is responsible for the COVID-19 pandemic, leading to a vast multitude of COVID-19 cases. Like all coronaviruses, SARS-CoV-2 viral RNA forms a functional structure referred to as a *pseudoknot* that, combined with a *slippery* RNA sequence, makes the ribosome prone to shifting into the −1 reading frame at the frameshift site [2–5]. Despite a high resolution of accuracy achieved for the study of specific RNA molecules, answering the question of how the frameshift event is regulated via the folding or unfolding of RNA structures remains elusive [6–9].

Given the high degree of complexity in predicting how the frameshift pseudoknot structure may be wedged or somehow possibly obstruct the entrance of the ribosome mRNA channel to initiate the frameshift event [10], we strive to fully understand the initial and subsequent folding of RNA molecules within a secondary structure model. To better inform and interpret tertiary structure experiments and simulations, our computational analysis explores the landscape of possibly pseudoknotted structures to elucidate key SARS-CoV-2 folding conformations. Experiments show that frameshift efficiency is significantly higher for an extended frameshift sequence (2924 nucleotides) than minimal frameshift sequence (92 nucleotides) [11]. Therefore, unfolding of longer range RNA structures must affect how the RNA refolds into a pseudoknot in proximity with the ribosome [12, 13]. Here we quest to further characterize the ensemble of most stable RNA secondary structures for extended frameshift sequences in SARS-CoV-2 and related coronaviruses.

Predicting and understanding frameshift inducing RNA structures in SARS-CoV-2 and related viruses is a critical target for therapeutic development [13, 14]. One example is intracellular small molecule therapy [10, 15–20], a strategy to limit viral fitness by disrupting the twisted RNA structure that contacts the ribosome at multiple locations [10]. Experiments demonstrate that specific compounds can inhibit frameshift efficiency in SARS-CoV-2 [10, 19] and other coronaviruses, affecting both humans and bats [21]. For example, KCB261770 was found to reduce frameshift efficiency in SARS-CoV-1, SARS-CoV-2, and MERS-CoV

[22]. Indeed, novel viral genome targeting found the SARS-CoV-2 frameshift pseudoknot site to be the most effective location to limit viral reproduction [23].

Vigilance is necessary as coronaviruses continue to evolve in animal reservoirs. SARS-CoV-2 viral evolution analysis finds it most phylogenetically related to bat SARS-like coronaviruses [24]. An intelligent strategy to prepare for future SARS-CoV-2 variants or novel coronaviruses must leverage evolutionary structure information as well as the *hierarchical folding hypothesis*, in order to understand the role of initial and subsequent RNA folding within frameshift element mechanics, e.g., our earlier work in predicting secondary structures for the 68 nucleotide (nt) SARS-CoV-2 sequence [25] (genome coordinates 13475−13542). Searching for commonality of structure features between coronaviruses contributes to broad spectrum pseudoknot therapeutic targeting, evidenced by molecular dynamic simulations of previously unknown 3-D structures for bat-coronavirus frameshift mechanics [26]. Previous analysis of length-dependent structures in different coronaviruses identified how sequences evolved to support a range of frameshift element structure motifs [13] (see Section 2.2).

Functional RNA structures that regulate the frameshift event have been studied for multiple viruses [27–29]. In particular, RNA sequences in the frameshift region have been observed to possess *conformational plasticity*, meaning they can form different configurations regulating frameshift efficiency [27]. For SARS-CoV-2, folding into multiple structures is a functional viral phenomenon evidenced by optical tweezers experiments [30, 31] as well as SHAPE-MaP (selective 2'-hydroxyl acylation analyzed by primer extension and mutational profiling) [32, 33] chemical probing of the SARS-CoV-2 viral genome *in vitro* [12] and *in vivo* [34–36]. Multiple unique stable conformations for the SARS-CoV-2 frameshift pseudoknot have been predicted and observed via various methods including crystallography, cryo-EM, and 3D-physics simulations [5, 12, 35, 37–39]. Thermal unfolding of RNA found major and minor paths from the folded to unfolded state, concluding that stability of transient states dictates folding paths [40].

Intense predictions efforts continue to build the structural model proposed for the exact SARS-CoV-2 −1 ribosomal frameshift mechanics. Mutations to the sequence of an RNA molecule can significantly disrupt structure-function. Analyses of mutated SARS-CoV-2 RNA sequences and resulting structures find even single nucleotide mutations can substantially reduce frameshift efficiency [10, 18, 41]. Empirical evidence supports the specificity needed for the SARS-CoV-2 frameshift sequence, i.e., that mutations in this structural region are remarkably rare. The most prevalent mutations in the frameshift region (C13536U and C13378U) were each observed in only 0.12% of over 700,000 sequences most recently recorded via GISAID at the time of this writing [42, 43]. Furthermore, there is some evidence of evolutionary convergence, with the most common mutation C13536U increasing sequence similarity between SARS-CoV-2 and MERS-CoV [44].

To contribute to comprehensive RNA structural knowledge we apply thermodynamic-based algorithms to predict the minimum free energy (MFE) *crossing* RNA structures (see Section 2.1) that may regulate ribosome pausing mechanics in betacoronaviruses. Towards resilient SARS-CoV-2 therapeutic treatments, we substantially expand on our previous hierarchical folding (i.e., non-crossing RNA structure forms first followed by more complex and possibly crossing base pairs [45–49]) investigation of the SARS-CoV-2 68 nt frameshift element sequence [25], by predicting secondary structures for extended length coronavirus frameshift sequences and incorporating sequence covariation information [50, 51].

We utilize MFE RNA *secondary structure* prediction algorithms, which output a set of base pairs, where each base is in at most one pair. Each RNA loop (i.e., unpaired region closed by a base pair) is assigned a free energy value. These energy values (Gibbs free energy) are either calculated empirically based on experiments such as optical melting [52] or are extrapolated

and estimated based on experimental results [53–55]. These energy values (referred to as energy parameters) are available for stacking base pairs varied by their sequence, structure initiation energies, and various types of mismatch penalties [56]. The set of these energy parameters is known as an energy model. The energy model we use in this work, that of HotKnots V2.0 [55] is based on temperature of 37˚C and 1 M salt concentration. RNA structure abstraction, such as secondary structure, enables efficient computational modeling while providing valuable structural information to help characterize RNA function. Predicting the tertiary (3D) structure of RNA [57–62] is crucial; however, it is more expensive computationally than our Shapify method ($\sim$ 3 seconds for structure prediction at sequence length 222 nt). 3D RNA models can be guided by secondary structure as an input constraint (for in-depth RNA tertiary structure prediction review, see Li et al. [63]).

To combat SARS-CoV-2 and prepare for novel coronavirus outbreaks, our strategy is to build and leverage coronaviruses structure-function information in support of frameshift-disrupting therapeutics. We focus on the secondary structure formation and function of the frameshift element viral RNA in betacoronaviruses, especially SARS-CoV-2. Specifically, we are interested in the most stable (i.e., MFE) initial stems, and subsequent hierarchical folding of viral RNA into possibly pseudoknotted secondary structures.

First, we sought to detect and utilize coronaviruses sequence covariation by applying *KnotAli* [64], a free energy minimization algorithm that merges conserved evolutionary information within a relaxed hierarchical folding approach. We utilized KnotAli to predict possibly pseudoknotted secondary structures for SARS-CoV-2 and related coronaviruses. We present and discuss our results for inter- and intra-coronavirus RNA structuredness. Predictions via KnotAli showcase evolved conformational flexibility based on detected covariation; they also provide insight on how mutations change predicted structures for SARS-CoV-2 and related viruses, especially bat coronaviruses. We provide additional context for known covariation and associated base pairing [37] by predicting novel base pairs that possess strong covariation within the multiple sequence alignment.

Second, to explore SARS-CoV-2 frameshift pseudoknot motifs in longer sequences, we applied our hierarchical folding algorithm, *Shapify*, to predict possibly pseudoknotted secondary structures while incorporating reactivity data. Following the approach of Schlick et al. [37], which extended to a 222 nt coronavirus frameshift sequence element window, we obtained SARS-CoV-2 frameshift element structure predictions for sequences increasing in length from 90 nt to 222 nt using Shapify, in combination with genome wide *in vivo* and sequence specific *in vitro* SHAPE data. Our predictions allow comparison of energetic stability between known secondary structure motifs [37]. Predictions via Shapify unveil a diverse array of energetically favourable potential pseudoknots for the frameshift element that were previously unknown. Our SHAPE-informed structure prediction analysis includes detailed classification of critical pseudoknot motifs. We report complex pseudoknot predictions both upstream (5′) and downstream (3′) with respect to the *native* pseudoknot structure (see Section 2.2), including the traditional *attenuator* hairpin [9]. By extending SHAPE-informed hierarchical-folding structure prediction, our analysis more precisely describes the landscape of energetically favourable RNA structures [65, 66], facilitating site-specific therapeutic targeting strategies.

## 2 Materials and methods

First, we introduce background information for RNA structure prediction and secondary structure motifs [25]. Second, we outline the hierarchical folding methods for secondary structure prediction. That is, we introduce KnotAli [64] for detection of covariation through

alignment and secondary structure prediction informed by covariation; and the Shapify algorithm [25] for hierarchical folding with SHAPE data as soft-constraint.

## 2.1 RNA secondary structure prediction

Computational methods to predict RNA secondary structure identify nucleotide bases that form base pairs when RNA molecules fold. RNA folding refers to the process by which RNA acquires its structure through base pairing. Prediction of the secondary structure (set of all base pairs) for an RNA molecule is a more achievable intermediate task shedding light on ultimately desirable RNA tertiary structure prediction. Thus, we focus on the prediction of RNA secondary structure, where any RNA molecule is represented by its sequence $S$ of length $n$.

Each RNA sequence has an alphabet of four bases: adenine (A), cytosine (C), guanine (G), and uracil (U). When an RNA structure forms, complementary bases pair together and form hydrogen bonds. *Canonical base pairs* are most stable pairings that occur when 'A' pairs with 'U', or 'G' pairs with either 'C' or 'U'. Each nucleotide base in the RNA sequence is referred to by its position in $S$ indexed from 1 to $n$ from 5′ (left) to 3′ (right) end. A *base pair* is defined as the pairing of two bases $i$ and $j$ where $1 \leq i < j \leq n$, and represented as $i.j$. Each base can pair with at most one other base (i.e., no base triplets are allowed, which is left to tertiary structure prediction). Consecutive base pairs are referred to as a *stem*.

The free energy of an RNA structure is calculated as the sum of the energies of its loops, i.e., hairpin, bulge, internal loop or multiloop. For this, some loop energies were experimentally determined, where experimental results were not available, loop energies are extrapolated [53–55]. The RNA structure with the minimum free energy (referred to as the *MFE* structure) is the structure with the lowest free energy for a given RNA sequence, and therefore, the most stable structure.

Base pairs $i.j$ and $i'.j'$ are *nested* if $1 \leq i < i' < j' < j \leq n$, and *disjoint* if $1 \leq i < j < i' < j' \leq n$. An RNA structure with only nested and disjoint base pairs is referred to as a *pseudoknot-free* structure. An RNA structure is considered *pseudoknotted* when at least two of its base pairs, $i.j$ and $i'.j'$ cross: $1 \leq i < i' < j < j' \leq n$, in which case both $i.j$ and $i'.j'$ are considered pseudoknotted base pairs.

In this work we assume RNA folds sequentially, supported by the *hierarchical folding hypothesis* [45]: *initial* stems first fold into a pseudoknot-free structure; subsequently, additional base pairing can lower the free energy of the structure and possibly form pseudoknots. Hierarchical folding paths were experimentally identified in multiple pseudoknots [48], including frameshifting pseudoknots [49]. Computational methods for prediction of RNA secondary structure based on the hierarchical folding hypothesis, including KnotAli [64] and Shapify [25], find the MFE structure for a given sequence $S$, and a pseudoknot-free input structure $G$.

RNA structure SHAPE chemistry experiments [46] and computational RNA folding simulations [47] find that initial structures may be modified locally to accommodate formation of more stable base pairs. Therefore, the KnotAli and Shapify algorithms are equipped to allow for minor modification to improve their prediction accuracy.

## 2.2 SARS-CoV-2 frameshift secondary structure motifs

SARS-CoV-2 secondary structure prediction efforts find structural *motifs* of several lengths related to the frameshift event [13, 37, 67]. We use *dual graph* nomenclature introduced in [37, 68–70] to refer to the RNA secondary structure or substructure predicted at or directly 3′ of the coronavirus frameshift element slippery sequence. A dual graph specifies connectivity and topological aspects of secondary structure by representing each stem by a vertex (see Fig 1).

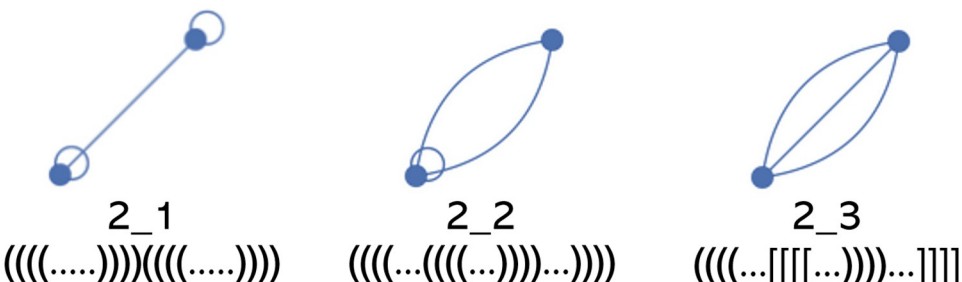

**Fig 1. RNA dual graph motifs and nomenclature with two vertices.** Vertices represent stems. Edges represent junction of stems, or bulge/internal loops with more than one residue on each strand. Self-edges represent hairpin loops. Dual graphs are referred to by two numbers, listed below each respective graph. The first number indicates the number of vertices, the second number specifies the topology, e.g., 2_1 is the dual graph secondary structure motif with two vertices, specifically the first possible topology. For additional details refer to the RNA-As-Graphs database [68]. Dot bracket example structures for each respective motif shown below number labels. An open parenthesis/bracket shows the opening base on the 5′ side of the base pair, a closed parenthesis/bracket represents the closing base on the 3′ side of the base pair. Each period "." represents an unpaired base.

Dual graph edges between vertices represent junction of stems, or loops. Specifically, each unpaired RNA strand is represented by an edge.

Dual graph representations allow for variable length of stems and loops, which makes RNA secondary structure pattern or motif identification within the ensemble of RNA structures easier. Note that the number of possible topologies rises exponentially with the number of stems, e.g., with three stems, there are eight possible topologies, with six stems, there are 508 possible topologies. Therefore, we use dot bracket notation (see Fig 1) or arc diagrams (see Fig 2) to represent secondary structure predictions with more than three stems, such as those predicted for the SARS-CoV-2 sequence of length 222 which may have ten or more stems. In arc diagrams, the RNA sequence of bases is shown from left to right (5′ to 3′) in a single horizontal line. Base pairs are represented as arcs connecting two bases.

The *native* frameshift element structure for SARS-CoV-2 [9], also referred to as the 3_6 motif [37], is a three-stemmed pseudoknot forming directly downstream of the slippery RNA sequence (see Fig 2 top arc diagram, pseudoknotted base pairs in red). Structure-function research has also identified a mechanism by which frameshift efficiency is downregulated.

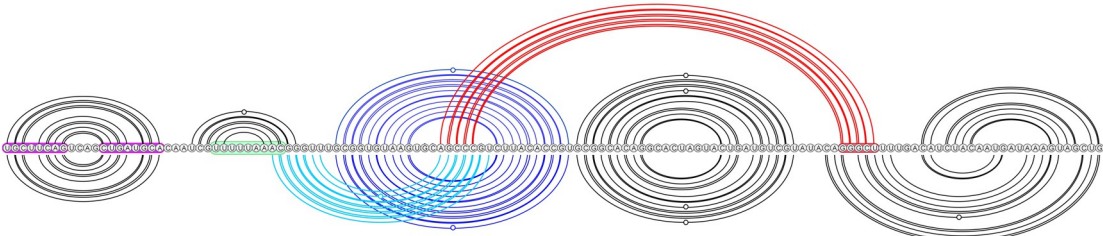

**Fig 2. Dominant SARS-CoV-2 pseudoknot motif predictions via Shapify.** SARS-CoV-2 frameshift element sequence shown as a horizontal line from 5′ (left) to 3′ (right). Arcs represent predicted base pairs. Top arc diagram includes 3_6 motif components (see Table 2 for dot-bracket format) of the fifth most stable structure predicted via Shapify (see Sections 2.4–2.6) for the 144 nt sequence (free energy −29.45 kcal/mol). Initial stem 5 base pairs in red (free energy −4.22 kcal/mol). Downstream pseudoknot target sequence highlighted in red. Bottom arc diagram includes 3_3 motif component of the MFE structure predicted via Shapify for the 144 nt sequence (free energy −30.93 kcal/mol). Initial stem 2 base pairs in light blue (free energy −6.1 kcal/mol). Initial stem 1 base pairs in dark blue (free energy −11.48 kcal/mol).

This involves a simple RNA loop located before the slippery sequence, known as the *attenuator* hairpin. The mechanism occurs through this loop's interaction with the ribosome during the translocation or elongation phases [9, 31] (see Fig 2, sequence highlighted in fuchsia). Previously, we predicted 3_6 motif structures and structure similarity with SARS-CoV-2 for SARS-CoV-1 and MERS-CoV [25].

Within a hierarchical folding framework for the SARS-CoV-2 frameshift element sequence, the stem occurring directly downstream of the slippery sequence, referred to as *stem* 1, was identified as the most energetically favourable initial stem (see Fig 2, dark blue arcs) within the 68 nt window [25]. Likely due to its high relative stability within the energy landscape, stem 1 refolds quickly when unfolded [30]. Stem 1 was also found to be highly conserved among coronaviruses [37]. It is present in both major structure motifs proposed by Schlick et al. [37] (3_6 for the 77 nt window, and 3_3 for the 144 nt window). For extended sequence lengths, native stem 1 may not form, instead, upstream base pairing has been identified [11].

Conversely, the second most energetically favourable initial stem for SARS-CoV-2 frameshift element sequence, i.e., stem 2, was found to pair differently depending on the window size around the frameshift site considered for prediction. Schlick et al. [37] used secondary structure prediction in combination with SHAPE structural probing and thermodynamic ensemble modeling to identify stem 2 either (*A*): paired into a simple pseudoknot crossing upstream of stem 1 in the 3_3 motif (95.6% of ensemble at 144 nt, see Fig 2, light blue arcs), or (*B*): paired downstream forming the native H-type pseudoknot in the 3_6 motif (98% of ensemble at 77 nt).

## 2.3 Covariation-informed hierarchical folding

KnotAli [64] combines the strengths of MFE RNA secondary structure prediction and alignment-based methods through relaxed hierarchical folding (see [71] for colloquial discussion of RNA folding history). KnotAli uses a multiple RNA sequence alignment as input to predict possibly pseudoknotted secondary structures for each sequence in the alignment. KnotAli first identifies a set of pseudoknot-free base pairs, based on mutual information, to guide subsequent free energy minimization. Note that relaxed hierarchical folding indicates predictions can be reached via multiple different paths allowing suboptimal structures, and the initial structure may be modified to accommodate base pairs that lower the free energy of the final structure. Output from KnotAli includes base pairs that show strong covariation among the multiple sequence alignment, as well as possibly pseudoknotted predictions for each individual sequence in the given multiple sequence alignment.

## 2.4 SHAPE-informed hierarchical folding

Shapify [25] is a relaxed hierarchical folding algorithm that uses as input an RNA sequence, a pseudoknot-free structure, and a SHAPE reactivity dataset, to predict a possibly pseudoknotted secondary structure for the given RNA sequence. Shapify follows four different biologically supported methods to predict the RNA secondary structure and returns the MFE structure from the four methods. In addition to the MFE structure we include any additional *suboptimal* structures resulting from the four methods that fall within 2 kcal/mol of the MFE structure in our results. Given the energy model used in our work (that of HotKnots V2.0 [55]) is based on temperature of 37˚C (310.15 K), a 2 kcal/mol temperature scale accepts minority populations down to 3.75% occupancy or roughly a difference of two or three base pairs.

To create the input structure following the hierarchical folding hypothesis, we used the HotSpots package (see HotKnots V2.0 [55]) and identified up to 20 lowest free energy unique

stems (referred to as *initial stems*) for each respective window to be used as constraint via Shapify. Assuming equi-probable RNA conformations, this cut-off is similar to the 3.75% occupancy threshold above. The stems were ranked from most stable to least stable based on their free energy and are referred to by their ranking as their IDs.

## 2.5 Coronaviruses alignment and sequence information

We obtained the coronavirus alignment of Schlick et al. [37], where out of 3760 SARS-CoV-2 coronavirus sequences [42], and 2855 other coronavirus sequences [72], 1248 sequences were found to be non-redundant [73]. These 1248 sequences were structurally aligned to the 222 nt SARS-CoV-2 frameshift element SHAPE consensus structure [37] using the Infernal covariance model [74] giving a final result of 182 non-duplicate homologous sites including seven SARS-CoV-2 sequences. Here, we converted the alignment of 182 sequences to FASTA format for input into KnotAli. The alignment is used as input to KnotAli to predict the secondary structure for the SARS-CoV-2 frameshift sequence.

We utilized the reference genome for SARS-CoV-2 from the National Center for Biotechnology Information, *NC*_045512.2 [75], as input to Shapify. We obtained three available SARS-CoV-2 SHAPE datasets to guide Shapify predictions, two *in vivo*: Huston et al. [34], and Yang et al. [36], one *in vitro*: Manfredonia et al. [35].

## 2.6 Shapify window procedure

Secondary structure predictions were obtained via Shapify with initial stems and SHAPE data for each SARS-CoV-2 sequence varying in length from 90 nt to 222 nt, with a step size of 1. Therefore, our results are based on removing successive nucleotides from the 5′ side of the RNA sequence for a total of 132 sequences analyzed (see Table 1 for window position and length). To compare free energy of structures for different sequence lengths, we divided the energy of each structure by its respective sequence length for an unbiased inter-window comparison, referred to as free energy per nt:

$$\text{Free energy per nt} = \frac{\text{Structure free energy}}{\text{Sequence length}} \qquad (1)$$

**Table 1. Shortest and longest window sizes used for SARS-CoV-2 structure predictions via Shapify.**

| Position | Length |
|---|---|
| 13485–13575 | 90 |
| 13354–13575 | 222 |

## 3 Results

We first visualize base pairs with strong covariation among the multiple sequence alignment identified by KnotAli. Next, we present secondary structure predictions for coronaviruses frameshift element sequences via KnotAli. Then, we present our structure exploration via Shapify by applying SHAPE reactivity data and following the hierarchical folding hypothesis to predict SARS-CoV-2 secondary structures for frameshift element sequences up to length 222. All predicted structures and associated data can be found in S1 File.

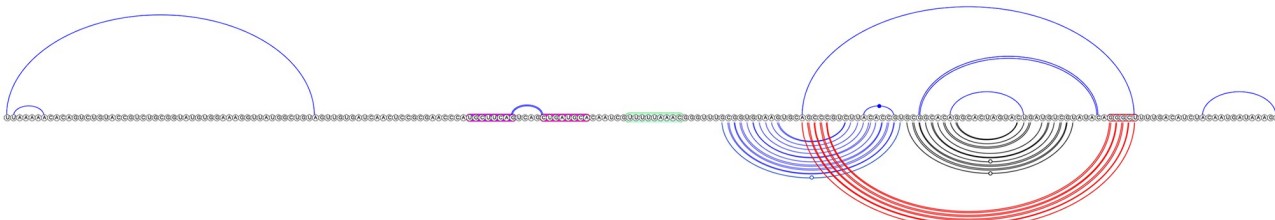

**Fig 3. Coronavirus frameshift element covariation.** Base pairs in the top arc diagram have strong covariation among the multiple sequence alignment identified by KnotAli. Bottom arc diagram displays the SARS-CoV-2 native 3_6 pseudoknot with the downstream target sequence in red. SARS-CoV-2 attenuator hairpin sequence highlighted in fuchsia and the slippery sequence in green.

## 3.1 KnotAli secondary structure predictions

KnotAli finds strong covariation in the multiple sequence alignment for base pairs that align with the known attenuator hairpin [9], as well as the native 3_6 motif (see Fig 3).

Among the seven SARS-CoV-2 sequences in the alignment, predicted structures for five sequences included the 3_3 motif, while two sequences (EPI_ISL_465643, EPI_ISL_426088) resulted in a prediction that included the native 3_6 motif instead (see Fig 4). A structure containing the 3_6 motif was also predicted for BtRf-BetaCoV (see Fig 5). BtRf-BetaCoV has 77% overall identity with SARS-CoV-2, and 91% frameshift sequence identity (FSID) with SARS-CoV-2 as identified by Schlick et al. [37].

The majority of sarbecovirus secondary structure predictions include the 3_3 motif: Pangolin-CoV (98% FSID), SARS-CoV-1 (93% FSID), SARS-like WIV1-CoV (93% FSID, see Fig 5), BtRs-BetaCoV (92% FSID), Bat-Cov-Cp (91% FSID), and Bat-CoV-Rp (91% FSID).

Sarbecovirus predictions demonstrate significant structuredness. Pseudoknots were predicted upstream of the native frameshift pseudoknot site for SARS-CoV-2, SARS-CoV-1, SARS-like WIV1-CoV, BtRs-BetaCoV, BtRf-BetaCoV, and Bat-CoV-Cp. Pseudoknots were predicted downstream of the native frameshift pseudoknot site for Pangolin-CoV, SARS-like WIV1-CoV, BtRf-BetaCoV, and Bat-HpBetaCoV.

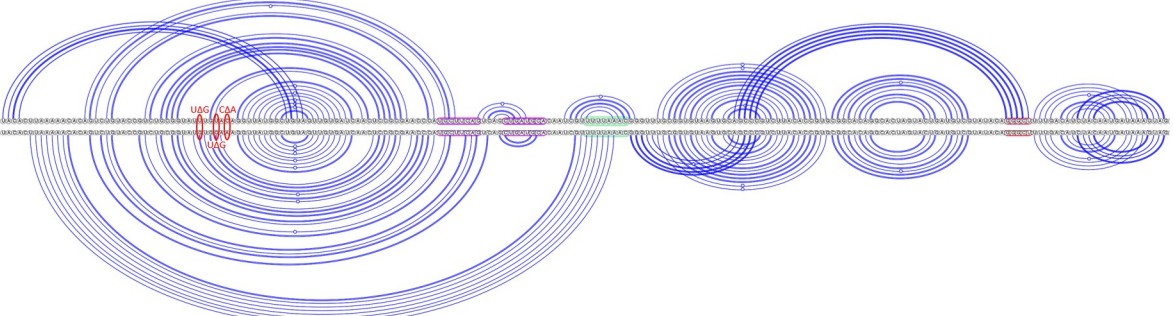

**Fig 4. SARS-CoV-2 secondary structure predictions via KnotAli.** Top arc diagram: free energy −36.47 kcal/mol, EPI_ISL_426088, includes 3_6 motif. Bottom arc diagram: free energy −40.65 kcal/mol, EPI_ISL_426905, includes 3_3 motif. Mutations are indicated with red ovals. A △ symbol represents the mutation from the nucleotide on the left to the nucleotide on the right. Attenuator hairpin sequence is highlighted in fuchsia, slippery sequence in green, and the downstream native pseudoknot target sequence in red.

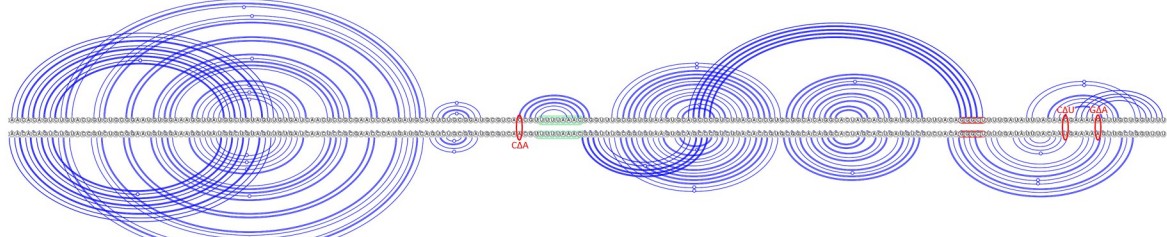

**Fig 5. Bat coronaviruses secondary structure predictions via KnotAli.** Top arc diagram: BtRf-BetaCov, free energy −43.24 kcal/mol, KJ473811, includes 3_6 motif. Bottom arc diagram: SARS-like WIV1-CoV, free energy −39.72 kcal/mol, KU444582, includes 3_3 motif. Mutations are indicated with red ovals. A △ symbol represents the mutation from the nucleotide on the left to the nucleotide on the right. Slippery sequence is highlighted in green and the downstream native pseudoknot target sequence in red.

### 3.2 Shapify secondary structure predictions

Following the procedure outlined in Section 2.6, we obtained a total of 10,916 secondary structure predictions via Shapify (see Fig 6) for the SARS-CoV-2 frameshift element window of varying length. Predictions via Shapify unveil changes in energetic favourability of pseudo-knotted motifs for extended length sequences. We observe that the most stable secondary substructures (including only up to three stems at or directly 3′ of the slippery sequence) can be classified into four pseudoknot dual-graph motifs: 2_3, 3_3, 3_6 (native-type), and 3_8 (see Figs 2 and 6, and Table 2).

Our predictions confirm that initial stem 1 folds into the 3_3 motif, as part of the complete MFE structure, for three specific window size intervals: 107−151, 191−199, and 216−222 (see Fig 6).

The 2_3 motif is energetically close to the 3_8 motif at window sizes 209−215. In addition, a 2_3 motif is predicted within the MFE structure for window sizes 205−208. We find the 3_8 motif predicted within the MFE structure for window sizes of 154−172 and 179−185. With window sizes 152−153, the MFE structure includes the 3_6 motif as a key component.

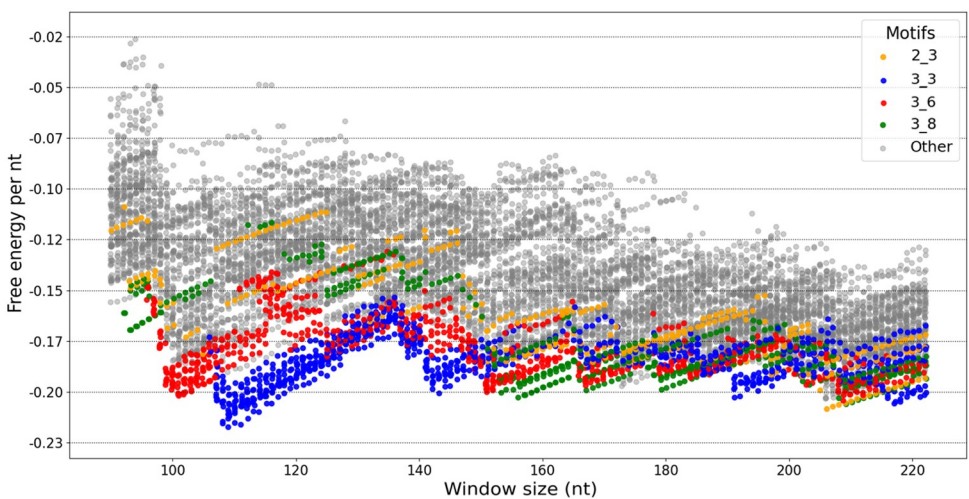

**Fig 6. SARS-CoV-2 secondary structure motifs free energy per nt.** Each dot represents a Shapify predicted secondary structure for the SARS-CoV-2 frameshift sequence (independent from KnotAli predictions, see Section 2.6, Table 1). X-axis represents window size, y-axis represents free energy per nt. Dots are colored based on the four listed dual-graph motifs (legend in top-right) detected at or directly 3′ of the slippery sequence (see Table 2), or grey for others. Noteworthy structures not matched with a motif (grey dots here) are visualized in Figs 8−10.

**Table 2. Secondary structure pseudoknot motifs in dot bracket notation.** Note that motif classification allows minimal modification to structures, e.g., in the size of loops. Sequence ID: *NC_045512.2* [75], indices 13467−13565. Open parentheses/brackets show the base on the 5′ side of the sequence, closed parentheses/brackets represent the base on the 3′ side of the sequence that are binding together. Each period "." represents an unpaired base.

| Motif | Secondary Structure |
|---|---|
| 2_3 | ..............................................................(((((((.[[.[[[.[[[[[)))))))........]]]]]..]]]]] |
| 3_3 | [[[[[[[.(((((((((((...]]]]]]])))))))))))(((((((((.........))).))))))................................................ |
| 3_6 | ........((((((((((.(([[[[[[)))))))))))))((((((((((....)).))).))))))......]]]]].............................. |
| 3_8 | ...............((((([[[[[[[.......[[[[[[[[[[[[)))))....]]].]]]]]]]]]]].]]]]]............................... |
| Sequence | ACGGGUUUGCGGUGUAAGUGCAGCCCGUCUUACACCGUGCGGCACAGGCACUAGUACUGAUGUCGUAUACAGGGCUUUUGACAUCUACAAUGAUAAAGUAGCUGGU |

There was path convergence, meaning multiple initial stems resulted in different predictions that each contain the 3_3 motif (see Fig 7). For example, with the 144 nt sequence as input, the MFE structure and the next four most stable structures all include the 3_3 motif.

At the critical window sizes 98−106, which include the region directly preceding the slippery sequence, the 3_3 motif is sufficiently destabilized leading to the MFE structure including the 3_6 motif instead. At the shortest length, as the 3_6 motif is destabilized, a 3_8 motif is predicted within the MFE structure for window sizes 93−97.

Beyond local pseudoknotted motifs, we detect additional stable pseudoknotted regions suggesting possible upstream and downstream structure-function of the frameshift element (see Figs 8, 9 and 10).

# 4 Discussion

Fully understanding functional RNA structures remains an elusive but worthwhile goal, and is a necessary step towards effective therapeutics for viral infections. Despite many attempts to distill how coronavirus RNAs fold to efficiently regulate frameshift events, much is still unknown about the mechanism. We employed two hierarchical-folding-based free energy minimization algorithms, KnotAli, and Shapify, for prediction of possibly pseudoknotted structures of SARS-CoV-2. Next, we discuss and contextualize our predictions of the

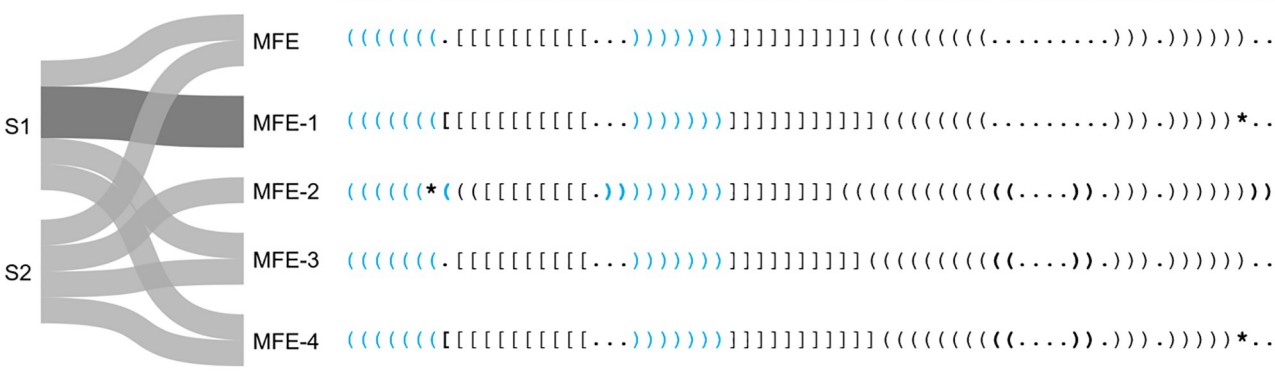

**Fig 7. Convergence to the most stable structures that contain the 3_3 motif.** 144 nt window Shapify predictions with initial stems 1 and 2 as constraint result in the MFE and most stable structures that contain the 3_3 motif. Initial stems are labeled on the left (e.g., S1 for initial stem 1). Darker grey path indicates the structure predicted with a specific initial stem was the same for two SHAPE datasets. Light grey path indicates the structure predicted with an initial stem was specific to one SHAPE dataset. Structures on the right are labeled by free energy proximity to the MFE structure. For example, MFE-1 is the lowest free energy structure after the MFE structure. Differences from the MFE structure are marked in bold, with parentheses representing changes in paired bases, and asterisks representing predicted unpaired bases that were paired in the MFE structure. The 3_3 motif pseudoknotted base pairs are shown in light blue.

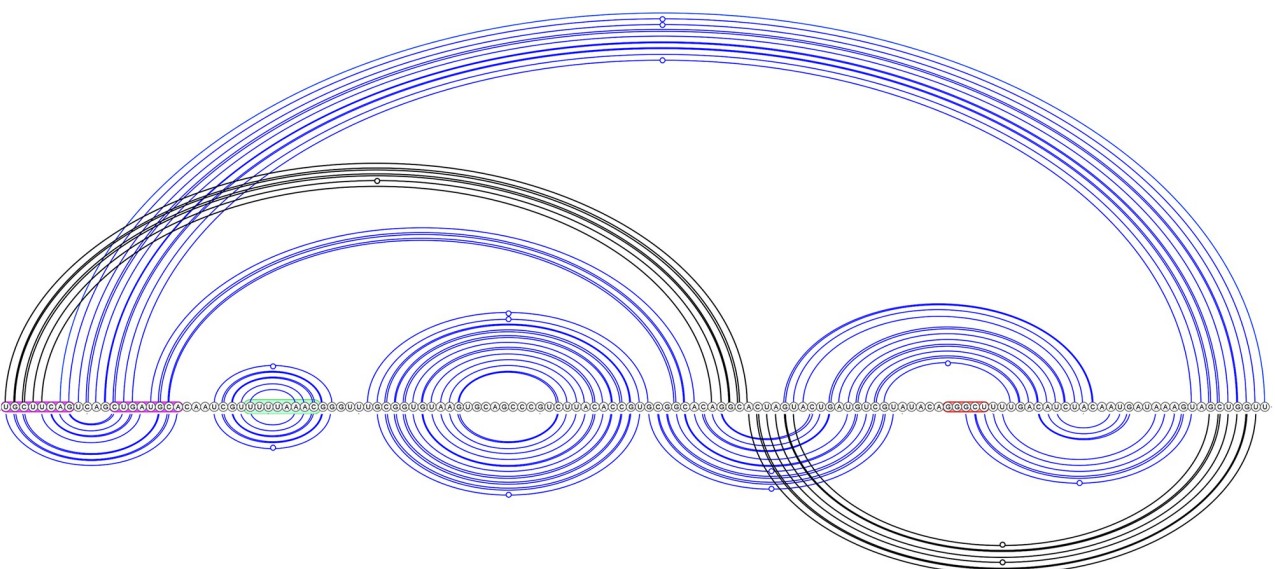

**Fig 8. Structural regions involving pseudoknots in SARS-CoV-2, 144 nt window via Shapify.** Top arc diagram: MFE-20, free energy −24.29 kcal/mol, initial stem 18 in black (free energy −0.35 kcal/mol). Bottom arc diagram: MFE-12, free energy −25.76 kcal/mol, initial stem 11 in black (free energy −1.44 kcal/mol). Attenuator hairpin sequence in fuchsia, slippery sequence in green, downstream native pseudoknot pairing region in red.

coronavirus frameshift element to advance known structure information towards site-specific viral therapeutics, namely: (1) secondary structure predictions for SARS-CoV-2 and bat coronaviruses frameshift elements via KnotAli, and (2) insights from SHAPE-informed hierarchical folding predictions for SARS-CoV-2 via Shapify.

## 4.1 Coronaviruses frameshift element covariation

In examining the base pairs identified by KnotAli that exhibit strong covariation within the multiple sequence alignment, we specifically highlight the innermost base pair of the traditional attenuator hairpin's stem-loop structure. This is a GC base pair located at positions 13442 and 13447. Covariation of this innermost base pair, together with the preservation of

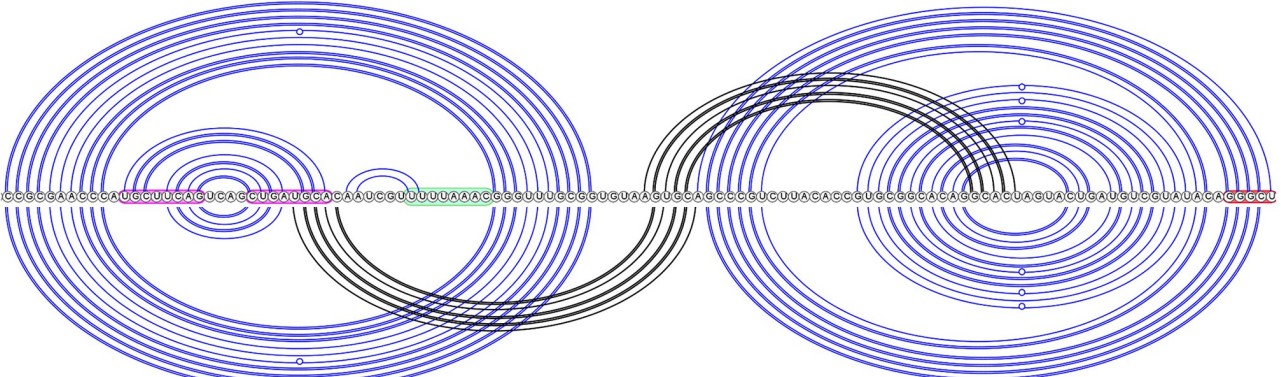

**Fig 9. SARS-CoV-2 pseudoknot predictions overlap, 222 nt window via Shapify.** Top arc diagram: MFE-5, free energy −45.04 kcal/mol, initial stem 15 in black (free energy −2.74 kcal/mol). Note that the MFE-5 pseudoknot was also detected within the 144 nt window (see MFE-19) and the 68 nt window [25]. Bottom arc diagram: MFE-29, free energy −39.34 kcal/mol, initial stem 12 in black (free energy −3.41 kcal/mol). Attenuator hairpin sequence in fuchsia, slippery sequence in green, downstream native pseudoknot pairing region in red.

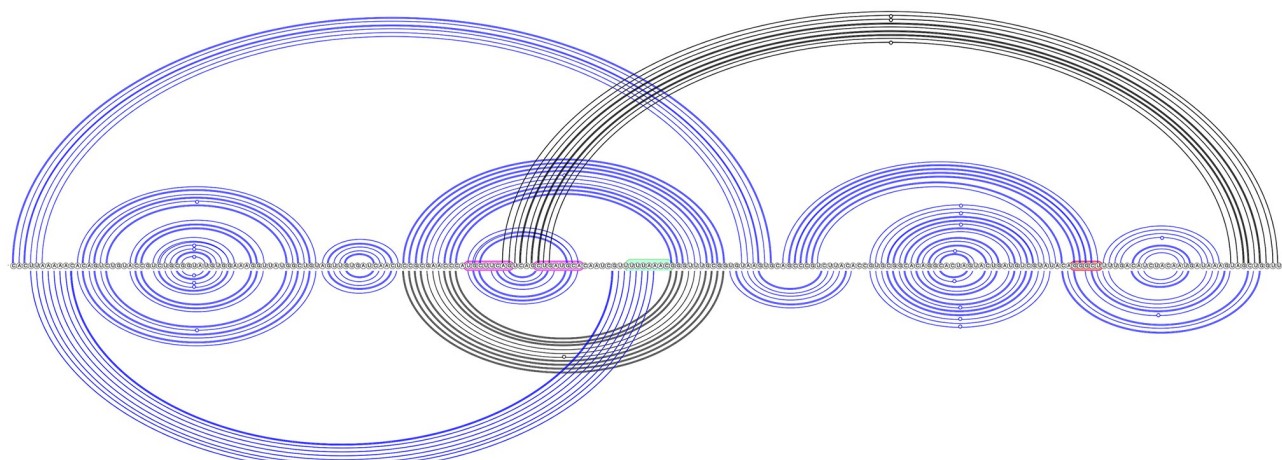

**Fig 10. SARS-CoV-2 long range pseudoknot predictions, 222 nt window via Shapify.** Top arc diagram: MFE-58, free energy −34.16 kcal/mol, initial stem 16 in black (free energy −1.98 kcal/mol). Bottom arc diagram: MFE-10, free energy −42.74 kcal/mol, initial stem 2 in black (free energy −10.08 kcal/mol). Attenuator hairpin sequence in fuchsia, slippery sequence in green, downstream native pseudoknot pairing region in red.

the attenuator hairpin's function in both SARS-CoV-1 and SARS-CoV-2 despite their sequence differences [9], suggests that the loop might be structurally conserved. Compensatory mutations related to the GC base pair at 13442.13447 include a potential AU base pair at 13442.13447 in transmissible gastroenteritis virus (TGEV, 59% FSID), porcine respiratory coronavirus (PRCV, 60% FSID), and turkey coronavirus (Turkey-CoV, 60% FSID), as well as a potential UA base pair at 13442.13447 in Night heron coronavirus (Night-Heron-CoV, 58% FSID). In addition, the outermost base pair of the pseudoknotted stem in the native 3_6 motif structure is found to have strong covariation, supporting previous results of Schlick et al. [37].

A two-branched multiloop directly downstream of the slippery sequence is identified by KnotAli as having strong covariation. The larger branch forms a bulge loop via a GC base pair at 13507.13536. Covariation of the GC base pair at 13507.13536 is supported by possible GU base pair at 13507.13536 in Pangolin-CoV, as well as a possible AU base pair at 13507.13536 in TGEV, Night-Heron-CoV, PRCV, Feline-CoV (61% FSID), and Canine-CoV (58% FSID). Further support is observed via possible UG base pair at 13507.13536 in infectious bronchitis virus (IBV, 60% FSID) and Turkey-CoV. Covariation of this bulge loop inner base pair, AU at 13512.13524, is supported by possible UA base pair at 13512.13524 in TGEV, PRCV, Feline-Cov, Canine-CoV and BtSk-Alpha-CoV (61% FSID), and also possible GU base pair at 13512.13524 in IBV and Turkey-CoV. In addition, the AU base pair at 13512.13524 is predicted via Shapify in multiple different SARS-CoV-2 pseudoknotted structures including the 3_3, 3_6, and 3_8 motifs (see Table 2) and longer range predictions (see Figs 9 and 10).

There are two base pairs with strong covariation predicted upstream, and one base pair with strong covariation predicted downstream of the native frameshift pseudoknot (see Fig 3). Previously, Schlick et al. [37] identified an upstream AU base pair at 13366.13411 in SARS-CoV-2, Pangolin-CoV, Bat-CoV-Rp, BtRs-BetaCoV, SARS-like WIV-1-CoV, SARS-CoV, BtRf-BetaCoV, and Bat-CoV-Cp. Their covariance analysis highlighted compensatory mutations via a CG base pair at 13366.13411 in TGEV and PRCV, a UG base pair at 13366.13411 in Feline-CoV, and a UA base pair at 13366.13411 in Rousettus-Bat-Cov (64% FSID). KnotAli identified strong covariation in the multiple sequence alignment to support a UA base pair at 13360.13366, with A13366 pairing upstream with U13360 as opposed to the previously identified AU base pair at 13366.13411. This UA base pair at 13360.13366 is supported by

compensatory mutations via possible GU base pair at 13360.13366 in TGEV, PRCV, Feline-CoV and Canine-Cov.

Finally, the additional upstream base pair predicted via KnotAli, UA at 13359.13410, is supported by possible GC base pair at 13359.13410 in IBV, Rousettus Bat-CoV, and Turkey-CoV. KnotAli also finds strong covariation in a downstream AU base pair at 13553.13565, supported by possible GC base pair at 13553.13565 in IBV and Turkey-CoV.

These novel predicted base pairings, supported by coronavirus multiple sequence alignment covariation, provide additional context to the previously identified covariation in coronaviruses [37]. We note the covariation-informed initial base pairs are preserved in certain individual predicted structures for coronaviruses, but in other cases more stable structures can be reached by disrupting these base pair(s). We observe that minimal upstream mutations to SARS-CoV-2 sequences lead to either the 3_3 motif or 3_6 motif predicted via KnotAli (see Fig 4). In addition, minor changes between bat coronaviruses like BtRf-BetaCov and SARS-like WIV1-Cov not only led to different motif predictions, but also affected predicted downstream pseudoknot pairings (see Fig 5). We conclude that minimal upstream and downstream sequence variation can significantly change conformations of the frameshift element. In supporting resilient frameshift therapeutics, it could be valuable to assess the viability of treatments intended for SARS-CoV-2 across various bat coronaviruses sharing similar sequences and structures. Such investigations could provide more detailed insights into how the frameshift regulation mechanism may be disrupted.

## 4.2 SHAPE-informed hierarchical folding

Our predicted secondary structures for the SARS-CoV-2 frameshift sequence via Shapify offer insight into frameshift dynamics. By considering a SARS-CoV-2 sequence of varying length as input to Shapify, we simulated how the interaction of the ribosome with the frameshift element [2] affects RNA structural motifs of SARS-CoV-2. In doing so we characterized the landscape of possibly pseudoknotted structures, finding the 3_8 motif to be the most energetically favourable at specific sequence lengths, while also clarifying interplay between the dominant 3_3 and 3_6 native-type −1 PRF structures.

Irrespective of length, structures containing the 3_3 motif reached the lowest free energy per nt (nearly −0.23, see Fig 6). Our SHAPE-informed hierarchical folding also demonstrates innate resiliency, i.e., redundant folding paths from different initial stems all leading to the 3_3 motif structure. This extends previous results finding similar path convergence to the 3_6 motif structure within a smaller 68 nt window [25].

Our results demonstrate that as shorter sequence length destabilizes the 3_3 motif, the 3_6 native-type motif seamlessly emerges to dominate the ensemble, suggesting a transition between the two motifs. We observe the shift from 3_3 to 3_6 as occurring at SARS-CoV-2 sequence index 13468, when the second of three successive guanine nucleotides that would otherwise form the 3_3 motif stem is removed from the sequence under consideration. This location for a potential transition of the secondary structure was identified within a constrained partition function framework for the SARS-CoV-2 frameshift element [76]. The structural ensemble of the SARS-CoV-2 frameshift element includes more than the 3_3 and 3_6 motifs. Of particular interest are the 3_8 and the 2_3 motifs.

The 3_8 motif was previously detected as part of the SARS-CoV-2 frameshift element by *in vivo* probing and subsequent folding analysis within a 126 nt window [34]. It was also predicted computationally via hierarchical folding informed by SHAPE reactivity data as soft-constraint within a 68 nt window [25]. Observing that the 3_8 motif is the most energetically favourable structure for shorter length sequences, we hypothesize it may act as a transient

structure facilitating refolding of either the 3_3 or 3_6 motif. In further support of a link between these three motifs is that they share an adenine bulge (A13524), which has long been identified as critical in frameshift regulation [77].

Further structure analysis is needed to understand function of the 2_3 motif, which is known to be prevalent in viral frameshift elements, especially plant viruses, but also simian retrovirus type-1, and mouse mammary tumor virus [70]. We hypothesize this downstream pseudoknot folding may have some effect on translation termination of the ribosome, which has been linked to frameshift regulation [78].

Our novel predictions suggest function of the attenuator hairpin via previously unknown pseudoknotted base pairing, paving the way for future tertiary modeling of SARS-CoV-2 frameshift RNA structure-function. Attenuator hairpin alternative pairings which include the slippery sequence and initial stem 1 were previously proposed [11]. Here, we predict for the first time attenuator hairpin bases folding into long range pseudoknotted interactions (see Fig 8, top arc diagram, and Fig 10, top arc diagram partially supported by IPKnot structure prediction [37, 79]), and also a potential H-type pseudoknot structure with the stem-loop conserved as part of a pseudoknotted multiloop (Fig 9, bottom arc diagram). Notably, we visualized multiple structures that possess significant overlap or regions of structural similarity. While certain stems overlap, other pseudoknotted base pairs form which indicate potential conformational switching between the two structures.

The latest experimental results on the tertiary structure of SARS-CoV-2 necessitated additional molecular folding simulations to achieve better alignment with the molecular structures derived from previous experiments [39]. Understanding how initial (most stable stems) give way to pseudoknotted motifs via unfolding and refolding may unlock the key to discerning kinetic trajectories, which are currently not well defined.

In general we find initial stem 1 as predicted by HotSpots [80], i.e, stem 1 (free energy −11.48 kcal/mol, see Fig 2, dark blue base pairs) to be the lowest free energy stem regardless of the window size. This confirms previous results finding initial stem 1 to be most stable within a 68 nt window [25]. Other initial stems vary in their relative energetic stability depending on the window size. For example, the initial stem 2 (free energy −6.1 kcal/mol, see Fig 2, light blue base pairs) is the second lowest free energy stem for the 144 nt window, but this is supplanted in the 222 nt window by an additional highly stable upstream stem. We hypothesize a possible frameshift downregulation function for this additional highly stable stem, as we found no folding path from this stem to the most stable secondary structures. Overall, further study is needed to fully understand the mechanics of the folding from initial stems into secondary structures, especially in pseudoknots.

In the folding process of the SARS-CoV-2 frameshift pseudoknot, the native pseudoknotted stem likely folds last [30]. Hence, we suggest exploring site-specific therapeutic targeting of the downstream native pseudoknot pairing. Since this pseudoknot critically refolds to initiate the frameshift, it might be an accessible location to disrupt the frameshift pseudoknot. In a comparison of target sites, it was found that an overlapping 5′ site, which excludes the pseudoknot target, was less effective than the downstream native pseudoknot pairing. The latter proved to be a more effective target for decreasing frameshift efficiency when targeted by oligonucleotides [81]. The downstream native pseudoknot site may be even more effective than previously realized, because it includes key structure pairings of the 3_8 and 2_3 motifs, which are each predicted within the MFE structure at specific sequence lengths. For more comprehensive and broadly applicable therapeutics, upstream sites, such as the attenuator hairpin, should also be considered in subsequent work. Further exploration of the intricate relationship between RNA structure and function is needed in the field of coronavirus therapeutics, towards more positive outcomes for human and animal health.

## Supporting information

**S1 File. Supplementary Materials.**
(ZIP)

## Acknowledgments

We thank and acknowledge the Computational Biology Research and Analytics Lab for invaluable feedback.

## Author Contributions

**Conceptualization:** Luke Trinity, Ulrike Stege, Hosna Jabbari.

**Data curation:** Luke Trinity.

**Formal analysis:** Luke Trinity.

**Funding acquisition:** Ulrike Stege, Hosna Jabbari.

**Investigation:** Luke Trinity.

**Methodology:** Luke Trinity, Ulrike Stege, Hosna Jabbari.

**Project administration:** Luke Trinity, Ulrike Stege, Hosna Jabbari.

**Software:** Luke Trinity.

**Supervision:** Ulrike Stege, Hosna Jabbari.

**Validation:** Luke Trinity.

**Visualization:** Luke Trinity.

**Writing – original draft:** Luke Trinity.

**Writing – review & editing:** Luke Trinity, Ulrike Stege, Hosna Jabbari.

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
