## [Decision Letter · Decision Letter 0]

29 Feb 2024

Dear Mr. Trinity,

Thank you very much for submitting your manuscript "Tying the Knot: Unraveling the Intricacies of the Coronavirus Frameshift Pseudoknot" for consideration at PLOS Computational Biology.

As with all papers reviewed by the journal, your manuscript was reviewed by members of the editorial board and by several independent reviewers. In light of the reviews (below this email), we would like to invite the resubmission of a significantly-revised version that takes into account the reviewers' comments.

We cannot make any decision about publication until we have seen the revised manuscript and your response to the reviewers' comments. Your revised manuscript is also likely to be sent to reviewers for further evaluation.

Sincerely,

Arne Elofsson

Section Editor

PLOS Computational Biology

Arne Elofsson

Section Editor

PLOS Computational Biology

Reviewer's Responses to Questions

**Comments to the Authors:**

Reviewer #1: The work of Trinity, Stege, and Jabbari in PCOMPBIOL-D-23-02112 describes a method of using phylogenetic covariation to constrain what are ultimately thermodynamic predictions of RNA conformations that are crucial to the mechanics of programmed ribosomal frameshift -1 RNA dynamics in coronaviruses, especially SARS-CoV-2. This is a key problem in CoV biology and pathogenesis and not understood well enough to permit the use of pharmaceuticals or even meaningfully constrain the evolutionary recombination dynamics in a class of viruses that undergo rampant combination. These are important questions, and this manuscript makes useful and probably important predictions of RNA conformation, key to this biology. It builds on earlier publications using these methods by the authors and others. The inclusion of sequence covariation is a very nice aspect of the paper and is appropriately central to their discussion.

The main proof of their method is the identification of new RNA conformations that are very likely to be important to the function of -1 programmed riboomal frameshifting.

Before publication, the authors should make a number of changes to clarify various points and clean up the manuscript in several areas.

1. This is large and growing field, and the authors do a reasonable job of citing the literature, but there are probably several citations to add. Please see below.

2. Clarify some language about the presence of multiple conformations in RNA.

3. Clarify some language about energy versus free energy of the system. Be more explicit about the implied temperature of the 2kcal/mol cut-off, being both 1000K or 3.4% occupancy at 298K. This analysis is unlikely to be sensitive to conformations with less than ~3% occupancy at room temperature.

4. Are the alignments used to calculate covariation available? If not, please state why not.

5. Please provide a rough estimate of the run time for methods described here. A detailed comparison with molecular dynamics is not needed, because I expect a massive speed-up, in addition to the inclusion of important information missing from MD, but some quantification of run time would be helpful.

Most of the bulleted items below are probably minor to fix, a few might be a bit of work. The starred items are more important.

Aspects of this work looks like it builds (at least indirectly) on ideas from Sean Eddy and colleagues 10.1093/bioinformatics/btaa080 and Wilson and colleagues https://pubmed.ncbi.nlm.nih.gov/10369773/, so cites to those two papers may be in order.

The last several sentences of the abstract should probably be shortened to the text in the Discussion: "Using a combination of thermodynamics methods and sequence covariation, our novel predictions suggest function of the attenuator hairpin via previously unknown pseudoknotted base pairing". Working in some version of, "While certain stems overlap, other pseudoknotted base pairs form which indicate potential conformational switching between the two structures." will nicely make the point that these sequence-informed analyses have an important message about the dynamics of -1PRF.

- Line 4: Number of covid infections is clearly billions, not hundreds of millions. Please adjust intro text. https://www.ncbi.nlm.nih.gov/pmc/articles/PMC9648705/

- Line 9: 'frameshift event is elusive' for citation to the physical basis for -1PRF. Please cite some subset of: https://www.ncbi.nlm.nih.gov/pmc/articles/PMC2670756/, https://pubmed.ncbi.nlm.nih.gov/18021801/, and/or https://pubmed.ncbi.nlm.nih.gov/25703095/, and the authors really should consider also using their citation #21 https://www.jbc.org/article/S0021-9258(17)50110-6/fulltext to be also cited here.

- Line 21: "predicting secondary structures for the 68 nucleotide (nt) SARS-CoV-2 sequence" Give the coordinates in the genome or in ORF1ab for the 68 nt structure in the text to clarify.

- line 57: Most or all large-ish biomolecule conformational states are transient, and intermediate states is the correct terminology, please remove the "i.e." and "transient" from the sentence.

*** Lines 60-62: "The RNA sequence itself is the 60 primary factor in determining the structure of an RNA, ergo mutations to the 61 frameshift element sequence can disrupt structure-function." This strong statement about RNA sequence should be moderated to give better context. Other physical effects aside from sequence play key roles in RNA structure. The fact that the authors are reporting a method that indirectly includes those non-sequence effects should be seen as progress for which they should be congratulated, but does not show other effects are nonexistent. https://www.ncbi.nlm.nih.gov/pmc/articles/PMC10091408/
https://pubs.acs.org/doi/epdf/10.1021/ja074191 etc. The rest of the paragraph is not invalidated or even weakened by correcting the first sentence in the paragraph.

- Line 74: Is the use of "cf." intended by the authors? https://www.law.cornell.edu/wex/cf Reading the text feels like they mean "i.e." or perhaps, "N.B."

*** Line 81: Please explain in more detail, "Each RNA loop (i.e., unpaired region 81 closed by a base pair) is assigned an energy value." What energy? The pair-binding energy? Is that energy only a function of the pair, and not the sequence context? Or is it assigned a free energy (i.e. including an assigned entropy)? Are this energies in the supporting information, or in the author's earlier publications?

*** Line 86: "it is more expensive computationally" I assume this means more expensive than the method presented in this paper. Please be explicit about the comparison of "more expensive". Than what? The speed of these calculations feels like it should be a significant advantage of this method over all-atom simulation, especially compared with atomistic solvent. Some rough quantification of the computation time of this method would be helpful.

- Line 88: "cf." is defined here: https://www.law.cornell.edu/wex/cf. I assume the authors mean "note well", or in Latin, "Nota Bene," "N.B."

- Line 121: Please cite https://pubmed.ncbi.nlm.nih.gov/20159159/
https://www.science.org/doi/abs/10.1126/science.1749933 for the text, "landscape of energetically favourable RNA structures," to clarify the standard but confusing terminology of "native pseudoknot". While it is clear that the authors understand the point about multiple conformations made by the two citations above, readers who are less familiar with physical chemistry and biological physics will benefit from the citations showing that "native state basin" is correct terminology while "native state" is wrong.

*** Line 161 "find the MFE (most stable) structure for a 161 given sequence S"

There is no requirement that the ground state of a system be unique (meaning in the limit of zero temperature). Degenerate ground states (where the objective function being minimize is the energy, not the free energy) are possible depending on system symmetry, including polymers and other glassy systems. A non-zero temperature, there can be many effectively degenerate ground states that -- in a coarse-grained/non-microcanonical model of polymer conformation as being used in this paper -- are best characterized by the system's entropy in a free energy of the system. I realize that the terminiology is "standard", but it is problematic. Please do what you can in your discussion to clarify.

- line 168 "SARS-CoV-2 secondary structure prediction efforts find multiple length-dependent 168 structural motifs related to the frameshift event"

I believe it would be clearer to write something like, "SARS-CoV-2 secondary structure prediction efforts find structural motifs of several lengths that related to the frameshift event"

- line 173: The use of "cf." is extraneous here. Translated from Latin, "cf." means "compare". Most likely the authors mean "q.v.". For further specifics, please see, https://writingcenter.unc.edu/tips-and-tools/latin-terms-and-abbreviations/

- Figure 2 caption: Do the authors mean the free energy, or the energy. Unless there is an implicit temperature and ion concentration in their calculations, then these are either (i) energies, not free energies, or (ii) the free energy is there, but possibly not completely well-defined as a computational expediency that allows for implicit inclusion as sequence information via a sequence alignment to known or predicted RNA structures. Later in the text, it is clearly free energy.

- Line 177: Is it possible that this text is making it unclear what is found in this analysis, in particular: windows lengths of RNA, versus what is actually present in the ensemble of RNA structures? If so, please clarify the wording.

- Line 214: Seems like there should be a way to work in a reference to https://cshperspectives.cshlp.org/content/10/10/a032433.full for "KnotAli [53] combines the strengths of MFE prediction and alignment-based methods 211 through relaxed hierarchical folding. KnotAli uses a multiple RNA sequence alignment 212 as input to predict possibly pseudoknotted secondary structures for each sequence in the 213 alignment." This history of RNA folding as been a long one, and citing a more colloquial discussion of that history will probably help the interested non-specialist reader.

*** Line 228: "With respect to the four structures from the four 226 methods, we include in our results any additional suboptimal structures that fall within 227 2 kcal/mol of the MFE prediction."

This is the first mention of a temperature scale I see in the paper. The free energy calculation are thus at 1000K, meaning that at 298K (room T), the calculation accepts minority populations down to 3.4% occupancy. Please spell this out. Here is an on-line converter: https://www.colby.edu/chemistry/PChem/Hartree.html

- Line 230: "and identified up to 20 lowest free energy unique stems" This implies a cut-off of 5% occupancy for roughly equi-probable conformations. For other distributions of occupancy/probability, it is unclear to me exactly what occupancy of the 20th state would be. But in any event, this is a heuristic that isn't too far off from the 3% occupancy above.

* Line 234: Section 2.5 Data. "We obtained the coronavirus alignment of Schlick et al. [33], where out of 3760 235 SARS-CoV-2 coronavirus sequences [38], and 2855 other coronavirus sequences [62], 236 1248 sequences were found to be non-redundant [63]. These 1248 sequences were 237 structurally aligned to the 222 nt SARS-CoV-2 frameshift element SHAPE consensus 238 structure [33] using the Infernal covariance model [64] giving a final result of 182 239 non-duplicate homologous sites including seven SARS-CoV-2 sequences."

This describes a major part of the methodology for why this approach is successful, and naming the subsection "Data" is an odd choice. The way the authors do their alignment here is a big part of the story: aligning to an RNA-world functional element. Please clarify the section headers. Is the alignment available as supporting information?

* Figure 4's red diamonds are essentially illegible. The figure should be re-made so that it can be studied and conveys the key information. No tiny symbols, please. The "pink" color looks purple to me.

* Figure 5. Same comments as for Fig. 4.

* Page 10/21: Please clearly state what was done with the conformations in Fig. 6 that are grey. Where they included or not included in further analysis leading to, for example figure 3 and 4 and 8-10?

* Figures 8, 9 , and 10: the different window lengths give rather different RNA conformations. The discussion of this section, "While 414 certain stems overlap, other pseudoknotted base pairs form which indicate potential 415 conformational switching between the two structures." is also one of the authors highlights.

*** Line 405: "Our novel predictions suggest function of the attenuator hairpin via previously 405 unknown pseudoknotted base pairing, paving the way for future tertiary modeling of 406 SARS-CoV-2 frameshift RNA structure-function." This is major message of the paper, but by my reading, this result is burried. Please repeat this text in a prominent location in the text.

- Line 419: "Understanding how initial most stable stems give way" Should be, "Understanding how inital (most stable stems) give way"

**Have the authors made all data and (if applicable) computational code underlying the findings in their manuscript fully available?**

The PLOS Data policy requires authors to make all data and code underlying the findings described in their manuscript fully available without restriction, with rare exception (please refer to the Data Availability Statement in the manuscript PDF file). The data and code should be provided as part of the manuscript or its supporting information, or deposited to a public repository. For example, in addition to summary statistics, the data points behind means, medians and variance measures should be available

---

## [Decision Letter · Decision Letter 1]

27 Apr 2024

Dear Mr. Trinity,

We are pleased to inform you that your manuscript 'Tying the Knot: Unraveling the Intricacies of the Coronavirus Frameshift Pseudoknot' has been provisionally accepted for publication in PLOS Computational Biology.

Best regards,

Arne Elofsson

Section Editor

PLOS Computational Biology

Arne Elofsson

Section Editor

PLOS Computational Biology

Reviewer's Responses to Questions

**Comments to the Authors: **

Reviewer #1: Trinity, Steve, and Jibarri have addressed the concerns and suggestions of this reviewer. The article is an important contribution to modeling the RNA conformational mechanisms underlying programmed ribosomal frame shifting and identified sequence-based signals reflecting those RNA-world effects.

**Have the authors made all data and (if applicable) computational code underlying the findings in their manuscript fully available?**

Reviewer #1: Yes

PLOS authors have the option to publish the peer review history of their article (what does this mean?). If published, this will include your full peer review and any attached files.

Reviewer #1: No

---

## [Editor Report · Acceptance letter]

3 May 2024

PCOMPBIOL-D-23-02112R1 

Tying the Knot: Unraveling the Intricacies of the Coronavirus Frameshift Pseudoknot

Dear Dr Trinity,

I am pleased to inform you that your manuscript has been formally accepted for publication in PLOS Computational Biology. Your manuscript is now with our production department and you will be notified of the publication date in due course.

With kind regards,

Judit Kozma
